mental health; MHPSS; natural disasters; developing country; climate change

**Corresponding author:**
Abhijit Nadkarni;
Email: abhijit.nadkarni@lshtm.ac.uk

# Barriers and facilitators to the implementation of mental health and psychosocial support programmes following natural disasters in developing countries: A systematic review

Olivia Rowe and Abhijit Nadkarni 

Centre for Global Mental Health, Department of Population Health, London School of Hygiene and Tropical Medicine, London, UK

## Abstract

Climate change is leading to more frequent and intense natural disasters, with developing countries particularly at risk. However, most research concerning mental health and natural disasters is based in high-income country settings. It is critically important to provide a mental health response to such events, given the negative psychosocial impacts they elicit. The aim of this systematic review is to explore the barriers and facilitators to implementing mental health and psychosocial support (MHPSS) following natural disasters in developing countries. Eight databases were searched for relevant quantitative and qualitative studies from developing countries. Only studies reporting barriers and/or facilitators to delivering MHPSS in response to natural disasters in a low- or middle-income country were included and full texts were critically appraised using the McGill University Mixed Methods Appraisal Tool. Reported barriers and facilitators were extracted and analysed thematically. Thirty-seven studies were included in the review, reflecting a range of natural disaster settings and developing countries. Barriers to implementing MHPSS included cultural relevance, resources for mental health, accessibility, disaster specific factors and mental health stigma. Facilitators identified included social support, cultural relevance and task-sharing approaches. A number of practical approaches can be used to facilitate the implementation of MHPSS in developing country settings. However, more research is needed on MHPSS in the developing country natural disaster context, especially in Africa, and international policies and guidelines need to be re-evaluated using a decolonial lens.

## Impact statement

This paper explores barriers and facilitators to delivering mental health and psychosocial support in developing countries following natural disasters. Given the vulnerability of the Global South to climate change, including increasingly frequent and intense natural disasters, this paper offers important learnings for both policy and programme delivery to build climate resilience and facilitate disaster response. It also addresses a key gap in the literature, which to date has predominantly focused on high-income country settings or in humanitarian contexts in general rather than specifically natural disasters.

## Background

Climate change is leading to more frequent and intense natural disasters in some regions of the world, with developing countries most at risk (Ludwig et al., 2007; IPCC, 2022). Developing countries are more exposed to the threat of natural disasters because of the geographical vulnerability of the global south in addition to the impact of disasters on existing challenges like poverty (IPCC, 2014).

Natural disasters are consistently associated with negative mental health impacts, including higher rates of psychological distress and mental disorders such as post-traumatic stress disorder (PTSD), depression, anxiety and suicidal ideation (Beaglehole et al., 2018; Cianconi et al., 2020; Palinkas and Wong, 2020). It is estimated that 25–50% of individuals impacted by natural disasters will experience negative mental health outcomes, with those living in developing countries more vulnerable due to increased exposure to natural disasters, increased levels of poverty and limited access to mental health services (Palinkas and Wong, 2020).

The risk factors that natural disasters pose to mental health can be both direct and indirect. Direct risk factors include exposure to the natural disaster itself, and indirect risk factors include

the impacts of natural disasters such as economic loss, poor physical health, displacement and civil conflict (Palinkas and Wong, 2020). These risks are further amplified in developing countries through the bidirectional relationship between poverty and mental illness (poverty is a risk factor for poor mental health and poor mental health is a risk factor for poverty) (Lund et al., 2011).

The worrying impact of natural disasters on mental health in developing countries sits against a backdrop of pre-existing, significant challenges concerning the prevalence, treatment and stigmatisation of mental disorders in developing countries (Horton, 2007). Low- and middle-income countries (LMICs) hold 75% of the global burden of mental illness, and an estimated 76–85% of those living with a mental disorder in LMICs do not receive treatment (Lopez et al., 2006; WHO, 2019a).

Mental health and psychosocial support (MHPSS), defined as 'any type of local or outside support that aims to protect or promote psychosocial well-being and/or prevent or treat mental disorder' are recommended by the United Nations (UN) Interagency Standing Committee (IASC) for implementation in response to emergencies, including natural disasters (IASC, 2007, p. 1). Most people will be able to recover from experiencing a disaster through basic MHPSS services like the provision of shelter, food and community support; however, a minority of individuals will require more focused or specialised care (DeWolfe, 2000).

MHPSS programmes, including basic services, community support and focused care, have been found to be effective in improving mental health outcomes in individuals affected by humanitarian emergencies in developing countries, including by improving psychological functioning and reducing the prevalence of PTSDs (Bangpan et al., 2019). However, there have been limited attempts to synthesise the literature on MHPSS delivery in response to natural disasters in developing countries. The existing literature often conflates natural disasters and conflict (Roudini et al., 2017; Troup et al., 2021). This is problematic because they are fundamentally different in nature with differing impacts on mental health (Altmaier, 2019), for example, the anger and paranoia following the Mumbai riots in 1992–1993 compared to persistent grief following the Indian Ocean tsunami (Makwana, 2019). Furthermore, different natural disasters may have different psychosocial impacts. For example, anxiety following flooding (Makwana, 2019) and psychological distress about radioactive materials following the Fukushima nuclear disaster (Harada et al., 2015). It is therefore reasonable to assume that the barriers and facilitators to the delivery of MHPSS in natural disaster settings may differ from that of other humanitarian settings, strengthening the rationale for the specificity of this review.

Furthermore, the majority of MHPSS research is conducted in higher-income country (HIC) settings despite in practice the overwhelming majority of MHPSS interventions taking place in LMIC humanitarian settings (Tol et al., 2011; Roudini et al., 2017). MHPSS programmes introduced in response to natural disasters can springboard transformational change in mental health systems. For example, the MHPSS response to the 2004 tsunami in Sri Lanka provided the impetus for a mental health system reform, which led to a significant scale up human resources for mental health and doubled the number of districts in Sri Lanka with mental health services infrastructure (WHO, 2022).

There are numerous challenges to delivering MHPSS which are particularly specific to a developing country context. The aim of this review is to understand the barriers and facilitators to the delivery of MHPSS programmes following natural disasters in developing countries. Understanding context-specific barriers and facilitators

to delivering MHPSS can offer useful insights for evidence-based policy making to address global mental health inequities.

## Methodology

### Design

Systematic review. The review protocol was registered a priori on PROSPERO (registration number: CRD42022348958).

### Eligibility criteria

*Population*: The target population includes individuals living in LMICs which had been indirectly or directly impacted by natural disasters or professionals who have delivered MHPSS support to said individuals.

*Intervention*: Only studies reporting barriers and/or facilitators to delivering MHPSS in response to natural disasters were included.

*Geographical location*: Only studies conducted in a LMIC were included.

*Study design*: Study designs included interviews/focus groups, cross-sectional studies, prospective studies, randomised controlled trials (RCTs), quasi-experimental studies, case studies and observational research.

*Language*: Only articles written in English or French were included in the review due to the linguistic competencies of the primary researcher.

*Date*: There were no date limitations for the research.

MHPSS programmes were defined in their broadest sense, covering basic services up to specialised psychological support (see Figure 1).

Developing countries were defined as LMICs as categorised by the *World Bank* for the most recent fiscal year (The World Bank Group, n.d.). The use of the term 'natural disaster' has been criticised by the academic community because it releases blame for disasters on manmade factors like preparedness policies and socioeconomic inequalities (Chmutina and von Meding, 2019). However, the term 'natural disasters' was used in this review due to its overwhelming use in the literature and because alternative phrases such as 'extreme weather events' exclude important disasters like earthquakes. Natural disasters were defined broadly as unavoidable environmental events that create fear of injury, loss of property and dislocation of residence and a wide range of search terms were used to reflect this (Altmaier, 2019). While the WHO defines mental health broadly as a state of wellbeing, for the purposes of a targeted search strategy, common mental disorders and their symptoms associated with natural disasters were focused on – depression, anxiety and PTSD. This led to a focus on MHPSS programmes with the explicit aim of alleviating the symptoms of mental disorders.

Barriers are defined as factors which prevent or impede the delivery of MHPSS, either through impacting the delivery of the intervention itself or the causal pathway between the intervention and its impact on mental health outcomes. Facilitators, on the other hand, are defined as factors which make the delivery of MHPSS easier.

### Search strategy

The three concepts of 'MHPSS programmes', 'developing countries' and 'natural disasters' were initially searched. The mitigation strategy for excessive results was to add the further concept of 'mental

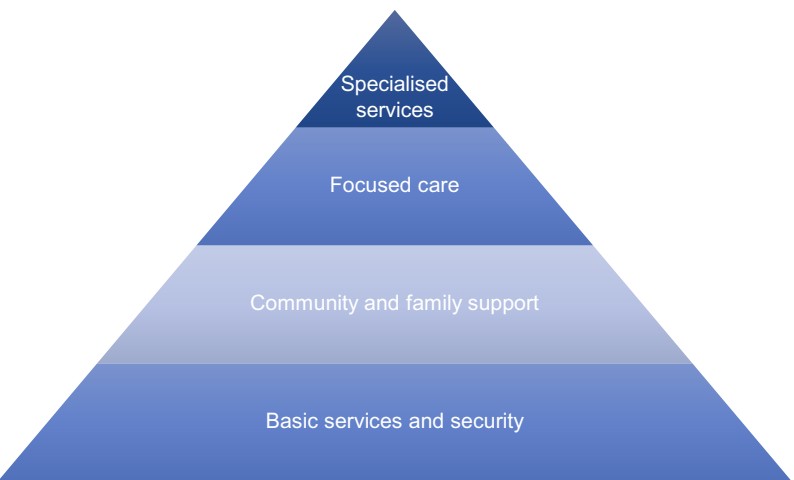

**Figure 1.** IASC intervention pyramid for MHPSS in emergencies (IASC, 2007).

health' and then 'disaster victim' if further narrowing of search results was needed. Detailed search strategy is described in Supplementary Appendices 1 and 2.

Embase, Medline, PsycInfo and Global Health databases were searched and the PRISMA[1] reporting guidelines were followed (Moher et al., 2009) (Appendix 3). Regional databases, African Journals Online, Latin American and Caribbean Health Sciences Literature, Nepal Journals Online and Sri Lanka Journals Online were also searched to capture more articles focused on developing countries and because Nepal and Sri Lanka are specifically very vulnerable to natural disasters (Eckstein et al., 2021).

Each database was searched in turn using the search strategy described above, duplicate studies were removed and final search returns were imported into the bibliographic software Mendeley. Titles and abstracts were screened according to the inclusion and exclusion criteria. Any studies that did not meet the inclusion criteria were excluded along with studies with unavailable English full text. Backward citation chaining was then conducted using references of included studies.

Data from the eligible studies were extracted into an extraction sheet designed to select data relevant to meet the objectives of the review (country and date of natural disaster; study design; sample; natural disaster; target mental health conditions; intervention; barriers; facilitators; Table 1).

### Analyses

A narrative synthesis was conducted in two stages in line with Bach-Mortensen and Verboom's (2020) recommendations on conducting systematic reviews of barriers and facilitators in health. Study characteristics were synthesised and descriptively summarised to generate a snapshot of the literature. Information on barriers and facilitators was analysed and each statement in the data extraction table was codified into a summary statement. The codes were then collated and categorised into broader themes, and the final results were summarised according to each theme.

Full texts were critically appraised using the McGill University Mixed Methods Appraisal Tool (Appendix 4) and no texts were excluded on this basis (Hong et al., 2018).

---

[1]PRISMA refers to Preferred Reporting Items for Systematic Reviews and Meta-Analyses.

### Results

Figure 2 describes the systematic review screening process. After screening out ineligible papers, 37 studies were eligible for inclusion in our review.

### *Study characteristics*

The included studies were cross-sectional surveys ($n = 11$), RCTs ($n = 7$), quasi-experimental studies ($n = 7$), interviews/focus groups ($n = 7$), mixed methods research ($n = 4$) and a case study ($n = 1$). Participants were almost all survivors of natural disasters ($n = 33$); with two studies each with a mixture of both survivors and humanitarian responders and only humanitarian responders. The majority of studies focused on the top layer of the MHPSS pyramid, specialised interventions or layer 4 ($n = 14$), followed by community and family support or layer 2 ($n = 9$), basic services or layer 1 ($n = 5$) and focused care interventions or layer 3 ($n = 5$), with the remaining studies incorporating a mix of MHPSS approaches ($n = 4$).

The country in focus was most often China ($n = 11$), followed by Haiti ($n = 7$) and Turkey ($n = 4$). Only two studies were conducted on the African continent, one in Burundi and one in Zimbabwe (Crombach and Siehl, 2018; Mhlanga et al., 2019).

The majority of natural disasters reported were earthquakes ($n = 24$) followed by tsunamis ($n = 6$). Other disasters reported on include hurricanes/cyclones ($n = 3$), flooding ($n = 2$), a mixture of flooding and tsunami ($n = 1$) and volcanic activity ($n = 1$).

A range of MHPSS programmes were implemented in the included studies. These included interventions at the higher end of the MHPSS pyramid like Narrative Exposure Therapy (NET) ($n = 7$) and Eye Movement Desensitisation and Reprocessing (EMDR) ($n = 2$), to more basic interventions like group psychoeducation and psychotherapy ($n = 7$), social work, for example, using social workers to connect different stakeholders and members of the community ($n = 2$) and shelter ($n = 2$).

### *Barriers to delivering MHPSS (Table 2)*

#### *Cultural relevance*

Some MHPSS programmes relied on western psychological practices, sometimes inappropriate to the specific developing country cultural context. For example, an MHPSS intervention for tsunami-

**Table 1.** Barriers and facilitators for implementation of MHPSS

| References | Country and date of natural disaster | Study design | Sample | Natural disaster | Target mental health conditions/symptoms | Intervention[a] | Barriers | Facilitators |
|---|---|---|---|---|---|---|---|---|
| Başoğlu et al. (2007) | Turkey, 1999 | RCT | Survivors | Earthquake | PTSD | Layer 4: Exposure therapy | Accessibility | |
| Başoğlu et al. (2005) | Turkey, 1999 | RCT | Survivors | Earthquake | PTSD | Layer 4: Cognitive Behavioural Therapy | | Task sharing |
| Becker (2009) | India, 2004 | Quasi-experimental | Survivors | Tsunami | General mental health symptoms | Layer 3: Training lay persons to provide psychosocial support | Shortage of mental health professionals | Task sharing. |
| Berliner et al. (2011) | Haiti, 2010 | Case study | Survivors | Earthquake | General mental health symptoms | Layer 1: Psychological First Aid | Non-local emergency responders, Disaster-specific factors | Normalisation of suffering, Social support |
| Catani et al. (2009) | Sri Lanka, 2004 | RCT | Survivors | Tsunami | PTSD | Layer 4: Exposure therapy and meditation-relaxation | | Social support, Task sharing |
| Chung (2017) | China, 2008 | Mixed method | Survivors | Earthquake | General mental health symptoms | Layer 2: Mix of psychoeducational groups | Accessibility, Short-term nature of response, Shortage of mental health professionals | |
| Crombach and Siehl (2018) | Burundi, 2014 | Quasi-experimental | Survivors | Flooding | PTSD, Depression | Layer 4: Narrative exposure therapy | Mental health stigma | Cultural relevance, Task sharing |
| Doocy et al. (2006) | Indonesia, 2004 | Mixed method | Survivors | Tsunami | General mental health symptoms | Layer 1: Cash for Work programme | Short-term nature of response, Non-local emergency responders | Social support |
| Farrell et al. (2011) | Pakistan, 2005 | Quasi-experimental | Survivors | Earthquake | Depression Anxiety | Layer 4: Eye Movement Desensitisation and Reprocessing (EMDR) | Short-term nature of response | Cultural relevance |
| Gao et al. (2013) | China, 2008 | Interviews | Survivors | earthquake | General mental health symptoms | Layers 2 and 3: Music therapy | Mental health stigma | |
| Gelkopf et al. (2008) | Sri Lanka, 2004 | Cross-sectional | Survivors | Tsunami | General mental health symptoms | Layer 3: Training lay persons to provide psychosocial support | Western psychological practices | Cultural relevance, Social support |
| Goenjian et al. (2021) | Armenia, 1988 | Prospective study | Survivors | Earthquake | PTSD, depression | Layer 3: Group psychotherapy | shortage of mental health professionals, Disaster-specific factors, Lost social networks | Social support |
| Ho et al. (2017) | China, 2008 | Cross-sectional | Survivors | Earthquake | anxiety | Layer 2: Art and play therapy | | Task sharing |
| Huang and Wong (2013) | China, 2008 | Focus groups | Survivors | Earthquake | General mental health symptoms | Layer 2: Social work (group recreational activities led by social workers) | Lack of prioritisation of mental health, Short-term nature of response | Social support |

(*Continued*)

| References | Country and date of natural disaster | Study design | Sample | Natural disaster | Target mental health conditions/symptoms | Intervention[a] | Barriers | Facilitators |
|---|---|---|---|---|---|---|---|---|
| James and Noel (2013) | Haiti, 2010 | Cross-sectional | Survivors | Earthquake | PTSD | Layer 3: Training lay persons to provide psychosocial support | Transitory lives of survivors | Task sharing |
| James et al. (2014) | Haiti, 2010 | Cross-sectional | Emergency responders | Earthquake | PTSD | Layer 2: Psychoeducation | Mental health stigma | Task sharing, Social support |
| James (2012) | Haiti, 2010 | Cross-sectional | Survivors | Earthquake | PTSD | Layer 3: Group psychotherapy | Disaster-specific factors | Task sharing |
| Jha et al. (2017) | Nepal, 2015 | Quasi-experimental | Survivors | Earthquake | PTSD | Layer 4: Cognitive Behavioural Therapy and Narrative Exposure Therapy | Lack of prioritisation of mental health | Task sharing |
| Jiang et al. (2014) | China, 2008 | RCT | Survivors | Earthquake | PTSD, depression | Layer 4: Interpersonal psychotherapy | | Social support, Task sharing |
| Konuk et al. (2006) | Turkey, 1999 | Quasi-experimental | Survivors | Earthquake | PTSD | Layer 4: Eye Movement Desensitisation and Reprocessing (EMDR) | | Task sharing |
| Krishnaswamy et al. (2012) | Malaysia, 2004 | Quasi-experimental | Survivors | Tsunami | Range of internationally recognised common mental disorders | Layer 4: Mix of psychotherapies like CBT | | Cultural relevance |
| Lane et al. (2016) | Haiti, 2010 | Cross-sectional | Survivors | Earthquake | PTSD | Layer 4: Narrative exposure therapy | | Task sharing |
| Leitch and Miller-Karas (2009) | China, 2008 | Cross-sectional | Survivors | Earthquake | General mental health symptoms | Layer 4: Trauma Resiliency Model | Shortage of mental health professionals | Cultural relevance |
| Madfis et al. (2010) | Haiti and the Solomon Islands, 2007 | Cross-sectional | Survivors | Flooding and Tsunami | General mental health symptoms | Layer 2: Safe Spaces (protection, education and psychosocial support to children) | Shortage of mental health professionals, Lack of prioritisation of mental health | Outreach activities, Cultural relevance |
| Meng et al. (2012) | China, 2008 | RCT | Survivors | Earthquake | PTSD | Layer 2: Chinese herbal medicine | Transitory lives of survivors | Cultural relevance |
| Mhlanga et al. (2019) | Zimbabwe, 2017 | Mixed method | Survivors | Cyclone | General mental health symptoms | Layers 1–3: Social work interventions including resource mobilisation, networking, counselling | Disaster-specific factors | Social support |
| Pérez-Sales et al. (2005) | El Salvador, 2001 | Interviews | Survivors | Earthquake | General mental health symptoms | Layer 1: Shelter Management | Lost social networks | Social support |
| Rodriguez-Sanjurjo (2021) | Puerto Rico, 2017 | Mixed Methods | Emergency responders/survivors | Hurricane | depression anxiety | Layer 1:Aid programme | Lost social networks | |
| Saint-Jean (2015) | Haiti, 2010 | Interviews/focus groups | Survivors | Earthquake | General mental health symptoms | Layer 2: Mix of MHPSS approaches including church initiatives, education | Short-term nature of response | Task sharing, Cultural relevance |

**Table 1.** (*Continued*)

| References | Country and date of natural disaster | Study design | Sample | Natural disaster | Target mental health conditions/symptoms | Intervention[a] | Barriers | Facilitators |
|---|---|---|---|---|---|---|---|---|
| Tasdik Hasan et al. (2020) | Bangladesh, 2007 and 2009 | Interviews | Survivors | Cyclone | PTSD Depression | Layer 1 and 4: Mix of MHPSS approaches including materialistic help and one psychiatrist | Shortage of mental health professionals, Accessibility, Mental health stigma, Lack of prioritisation of mental health | Task sharing, Cultural relevance |
| Vijayakumar and Kumar (2008) | India, 2004 | Cross-sectional | Survivors | Tsunami | PTSD, Depression | Layer 2: Emotional support via befriending | | Cultural relevance, Task sharing |
| Wolmer et al. (2005) | Turkey, 1999 | Quasi-experimental | Survivors | Earthquake | General mental health symptoms | Layer 2: School-based intervention | Short-term nature of response | Task sharing |
| Wu et al. (2019) | China, 2016 | Interviews | Emergency responders | Flooding | Not reported | Layer 1: Shelter | Lack of prioritisation of mental health, shortage of mental health professionals, Mental health stigma | |
| Xu and Deng (2013) | China, 2008 | Interviews | Survivors | Earthquake | PTSD | Layer 4: Professional mental health services | Accessibility | Outreach activities |
| Zahlawi et al. (2019) | Vanuatu, 2017 | Cross-sectional | Survivors | Volcanic activity | General mental health symptoms | Layer 3 and 4: Mix of MHPSS approaches including professional and traditional community | | Cultural relevance, Task sharing |
| Zang et al. (2013) | China, 2008 | RCT | Survivors | Earthquake | PTSD, Depression, anxiety | Layer 4: Narrative Exposure Therapy | Transitory lives of survivors | |
| Zang et al. (2014) | China, 2008 | RCT | Survivors | Earthquake | PTSD | Layer 4: Narrative Exposure Therapy | Accessibility | Social support |

[a]MHPSS: Layer 1, basic services; Layer 2, community and family support; Layer 3, focused care; Layer 4, specialised services (United Nations Children's Fund, 2022).

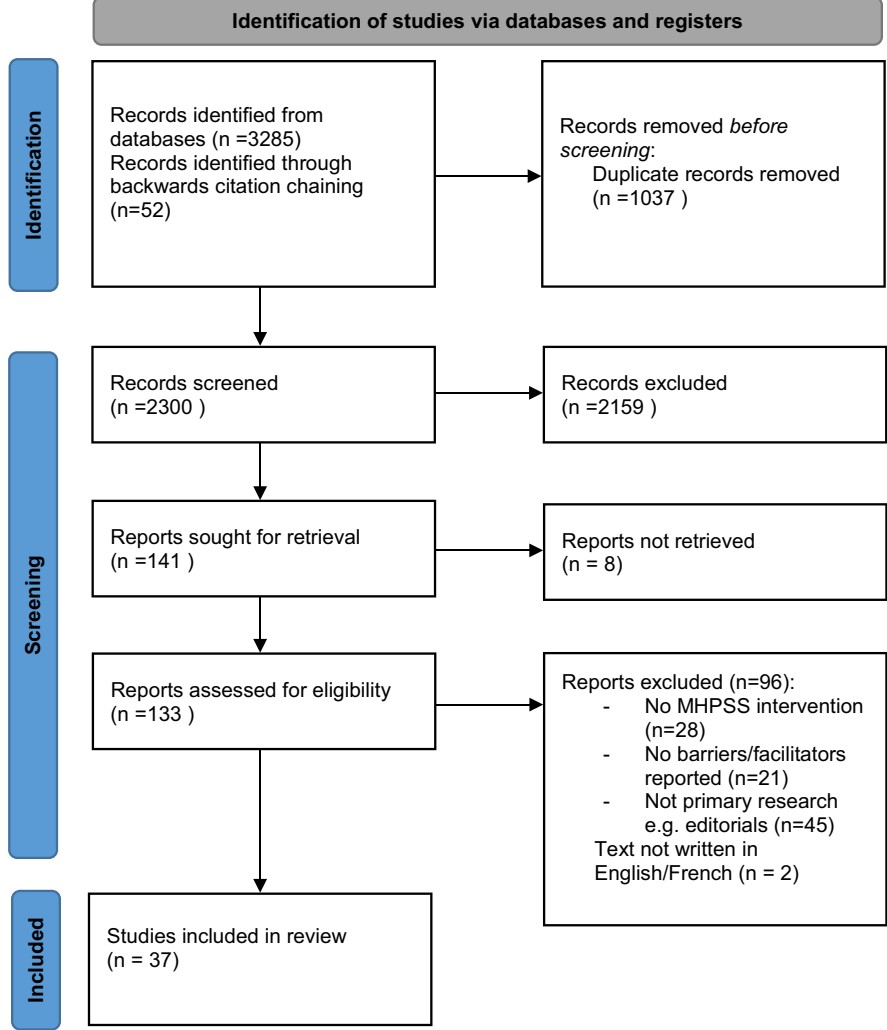

**Figure 2.** PRISMA flow diagram.

**Table 2.** Overview of barriers and facilitators

| Barriers | Facilitators |
| --- | --- |
| Cultural relevance | Social support |
| Resources for mental health | Cultural relevance |
| Accessibility | Task-sharing approaches |
| Stigma | |
| Disaster specific barriers | |

survivor children in Sri Lanka found that Western models of cognitive behavioural treatment (CBT) were found to be inappropriate in the local cultural context (Gelkopf et al., 2008). This was because CBT practices like self-affirmations and challenging negative thoughts were looked upon in Sri Lankan culture as a sign of weakness and against the Buddhist belief that trauma and distress should be accepted as a natural experience.

Furthermore, non-local MHPSS staff were reported to lack understanding of the local cultural context. Doocy et al. (2006) analysis of a Cash for Work programme response to the tsunami in Indonesia reported that programmes delivered by organisations without previous experience of working in the area had difficulty understanding the local culture and the communities' needs.

### Resources for mental health

The immediate focus following disaster tended to be on basic necessities like food and shelter with authorities tending to neglect MHPSS interventions (Wu et al., 2019; Tasdik Hasan et al., 2020). Some studies also talked about the importance of prioritising social work for mental health. For example, in Mhlanga et al.'s (2019) study of social work interventions for natural disasters in Zimbabwe, it was reported that there was a critical lack of child protection services in affected districts with significant exposure to verbal and sexual abuse.

A shortage of qualified mental health professionals was consistently reported as a barrier. Leitch and Miller-Karas (2009) reported that as a result of the earthquake in China, many workers were killed or injured, so the number of workers left had very little capacity to engage in MHPSS due to the need to share more workplace responsibilities. Madfis et al. (2010) found the lack of personnel meant it was challenging to provide differentiated support to different age groups of children. The misconception behind this barrier is that only qualified mental health workers are able to

provide MHPSS, when the finding around task sharing described below demonstrates that this is certainly not the case.

Even if resource was dedicated to an MHPSS response, this was often criticised as unsustainable with external mental health professionals (Doocy et al., 2006; Saint-Jean, 2015). Jha et al. (2017) in their study of earthquake survivors with posttraumatic stress in Nepal reported that as the humanitarian mental health response closed, the responsibility of addressing mental health needs was passed to Nepal's Ministry of Health where no plans were made to support the long-term rehabilitation of survivors.

### Accessibility

Accessibility barriers included reports that mental health services were difficult to reach due to limited transport, money or time (Tasdik Hasan et al., 2020). Other studies reported accessibility barriers for particular populations. For example, Madfis et al.'s (2010) emergency safe spaces intervention for children affected by disaster in Haiti and the Solomon Islands commented that disabled children, girls and minority language and ethnic groups participated less in the programme. The lack of access to services for disabled persons was highlighted in Chung's study of a post-earthquake rehabilitation programme in China, due to an unequal power dynamic between health care professionals and programme users and/or participants' limited understanding of their rights to participate (Chung, 2017). Several studies also reported that men were less likely to access the MHPSS intervention because the time of the intervention was during hours when men were typically out for work (Başoğlu et al., 2007; Zang et al., 2014).

### Stigma

Stigma was a frequent barrier to engaging with MHPSS, often exacerbated by poor community knowledge about mental health. For example, Wu et al.'s (2019) study of shelter responses to flooding in China detailed mental health stigma in China, quoting a folk proverb 'Prevent fires, deter thieves and be wary of psychologists'. Furthermore, one study interviewing cyclone survivors in Bangladesh reported one respondent who said going to hospital for mental health reasons was shameful (Tasdik Hasan et al., 2020). This stigmatisation of mental health prevents the update of MHPSS services following a natural disaster and is therefore a critical barrier.

### Disaster specific barriers

Sometimes the intervention itself was not physically possible due to the weather (James and Noel, 2013). Furthermore, the long-term nature of some natural disasters like earthquakes, with aftershocks sometimes occurring for months after the initial event, led to continuous fear. Berliner et al.'s (2011) case study of an earthquake survivor in Haiti reported that the individual was constantly afraid of being in the hospital in case another earthquake destroyed the building. However, some studies revealed the challenge of separating the distressing impact of natural versus man-made disasters. For example, in Farrell et al.'s (2011) study of an EMDR intervention following the 2005 earthquake in Northern Pakistan, very few of the participants' distress was focused on the earthquake itself with other factors like terrorism and domestic violence often being the main sources of concern.

Studies also reported delivery barriers to the MHPSS intervention because of the transitory nature of disaster survivors' lives, making longer-term follow up challenging (Başoğlu et al., 2005; Meng et al., 2012; Zang et al., 2013). For example, Meng XianZe and

colleagues found that 45% of participants in their RCT screened for PTSD were lost to follow up (Meng et al., 2012).

### Facilitators to delivering MHPSS (Table 2)

#### Social support

Pérez-Sales et al. (2005) research on shelter management after earthquakes in El Salvador compared shelter which grouped tents in order of arrival with another shelter which grouped tents to reflect the original communities of the survivors. In the camp organised according to communities, outcomes such as mental health symptoms and participation in camp activities were superior compared to the comparator. Some studies found explicitly improving social support as part of the MHPSS intervention led to positive mental health outcomes (Doocy et al., 2006; Jiang et al., 2014; Zang et al., 2014). For example, Jiang and colleagues argued that interpersonal psychotherapy was effective in improving mental health because it helped to reconstitute social support which is an essential element of PTSD recovery (Jiang et al., 2014). MHPSS for relatives of survivors was also found to strengthen their ability to support the survivor in question, and training lay people as mental health workers reportedly improved the volunteers' own mental health through the social connections they were able to build (Berliner et al., 2011; James et al., 2014).

#### Cultural relevance

Cultural relevance was seen as an important facilitator of effective MHPSS delivery. For example, Saint-Jean and colleagues examined the experiences of Haitian earthquake survivors in relation to the MHPSS disaster response and reported cultural relevance as a facilitator, achieved through local participation in committees (Saint-Jean, 2015). Furthermore, an RCT based in China explored the use of traditional Chinese herbal medicine, a widely used and culturally accepted practice, to improve the mental health of earthquake survivors (Meng et al., 2012). The study found that traditional medicine is cheap and quick to distribute, and is associated with significantly improved mental health symptoms compared to a controlled placebo.

One intervention in post-earthquake China sought to overcome cultural barriers to Western psychological practices by focusing at the biological level instead of using a form of psychotherapy called the trauma resiliency model which focuses on the biological impacts of traumatic symptoms on the nervous system (Leitch and Miller-Karas, 2009). In this study, almost all healthcare workers surveys indicated this biological model of trauma would be useful for their work with earthquake survivors.

The positive impact of culturally relevant MHPSS was also seen for programmes that were able to adapt based on an awareness of local cultural contexts, including mental health stigma and marginalised populations. For example, Gao's study of music therapy for Sichuan earthquake survivors reported that music therapists deliberately avoided the word 'therapy' to increase engagement (Gao et al., 2013). Crombach and Siehl's (2018) study of natural disasters in Burundi found that the use of local counsellors helped to combat mental health stigma. Furthermore, Berliner's study of Haitian earthquake survivors found that the normalisation of suffering was seen to tackle prominent mental health stigma (Berliner et al., 2011). Interestingly, Xu and Deng's (2013) study of mental health service use in post-earthquake China reported that outreach activities like home visits as well as low service charges for lower income groups could support accessibility of MHPSS.

### Task-sharing approaches

The most frequently reported facilitators of MHPSS interventions were task-sharing approaches, often called 'train-the-trainer' interventions (*n* = 14). This method aims to deliver MHPSS in a resource-poor environment and overcome cultural barriers by training laypersons in the community in basic psychotherapy and psychoeducation practices to provide MHPSS to their communities. Lane et al.'s (2016) study of the task sharing approach for delivering trauma therapy to Haitian earthquake survivors described the model as a 'force multiplier' in areas with significant mental health needs and insignificant professional resources. Zahlawi's survey of survivors of volcanic activity in Vanuatu found that a significant minority of respondents (18%) used traditional and community networks as their only source of psychosocial support as opposed to more professional MHPSS, suggesting that the train the trainer model could be an important facilitator of MHPSS in areas where individuals either cannot or do not want to access professional services (Zahlawi et al., 2019). The model was reported as beneficial for the community trainers' own mental health, with one intervention in post-earthquake Haiti associated with decreased PTSD symptoms, significant posttraumatic growth and positive qualitative accounts from trainers about their experiences delivering MHPSS (James et al., 2014). Task-sharing interventions were also suitable for overcoming some of the barriers to accessibility, such as language barriers.

## Discussion

This review has highlighted a range of natural disaster settings and developing countries, with China (*n* = 11) and earthquakes (*n* = 24) being the most common areas of focus. Barriers to implementing the MHPSS included cultural relevance, resources for mental health, accessibility, disaster specific factors and mental health stigma. Facilitators identified included social support, cultural relevance and task-sharing approaches.

Our biggest overarching finding is the need for an in-depth understanding of the local sociocultural, political and economic context in order to sensitively adapt an MHPSS intervention to maximise effectiveness.

The importance of cultural relevance mirrors the existing IASC guidelines on MHPSS delivery in emergency settings (IASC, 2007). However, what is perhaps missing from current IASC guidance is the relevance of western psychological approaches to disaster settings in LMICs. It may not be sufficient to understand the local context if the MHPSS you are using is founded upon the western cultural context and not adapted to the culture in which it is being implemented, such as CBT with an individualistic focus which has not been adapted for use in a more collectivist culture (Gelkopf et al., 2008).

This finding relates to an issue recently brought to light in the global mental health discourse; decolonialising global mental health (Weine, 2021). The evidence base for WHO treatment guidelines for mental health care are founded upon western psychological practice (Horton, 2007; Mills, 2014). While there is increasing recognition of the effectiveness of alternative non-Western practices like yoga on mental health, the evidence base, particularly for an LMIC disaster setting, is arguably yet to fully emerge (Kirkwood et al., 2005; Butterfield et al., 2017; Weine et al., 2020). Examples like Meng et al.'s (2012) Chinese herbal medicine to support Sichuan earthquake survivors with PTSD highlight an emerging evidence base for future research to build on. The implications for disaster policy include building in local participation in MHPSS disaster preparedness and response planning, and looking beyond western research and international guidelines to the local evidence base on MHPSS to inform policies. Ensuring the availability of quality mental health services and community support structures in advance of natural disasters is an important part of disaster preparedness and a key mechanism to empower culturally relevant community responses instead of relying on non-local emergency responders. Indeed, this review highlights barriers regarding resources and personnel for mental health, which corroborate the stark known mental health treatment gaps in LMICs (WHO, 2019a).

This review also adds to the strong existing evidence base on task sharing in resource-poor contexts and particularly communities where mental health stigma is rife (Padmanathan and De Silva, 2013; Hoeft et al., 2018). Mental health stigma is a global phenomenon, illustrating that lessons learned from mental health approaches in LMICs should not be siloed to the developing world (Lasalvia et al., 2013). Nevertheless, this review demonstrates that the stigmatisation of mental health continues to be a significant challenge in developing countries. This finding supports the IASC guidelines recommendation to 'implement strategies for reducing discrimination and stigma of people with mental illness' (IASC, 2007). The continued criticality of global mental health stigma has been recently highlighted in the Lancet commission to end stigma and discrimination in mental health, co-produced with persons with lived experience (Lancet, 2022). This review has highlighted specific programmatic and policy approaches to achieve this, such as psychoeducation and the normalisation of mental health responses to natural disasters.

A barrier to many MHPSS interventions included in this review was accessibility. Our findings support emerging research that disabled individuals in particular are often left behind in responses to disasters worldwide (Quaill et al., 2018). Implications for MHPSS practice include the need for proactive engagement with hard-to-reach communities. This is in line with IASC guidelines which place human rights and equity as the first principle of delivering MHPSS in emergency settings (IASC, 2007). Furthermore, this finding supports the launch of the WHO QualityRights Initiative in 2019, a comprehensive training package which aims to promote a human rights approach in the area of mental health, including the rights of persons with disabilities to access quality mental health services, in line with the UN Convention on the Rights of Persons with Disabilities (WHO, 2019b). Future research on MHPSS responses to disaster in LMICs needs to disaggregate results by different population groups in order to build the evidence base on which groups are left behind, and methods for improving accessibility. Policies should also reflect vulnerable groups to be targeted in MHPSS outreach activities. For example, Indonesia's disaster management policy specifies pregnant women and children as vulnerable groups for psychosocial services (Law of the Republic of Indonesia Number 24 of 2007 Concerning Disaster Management, 2007).

Social support is frequently reported in the wider literature as a critical factor supporting mental health and wellbeing (Fasihi Harandi et al., 2017). Furthermore, recent research on the role of social cohesion and community resilience in the context of the COVID-19 pandemic, demonstrates that strong social cohesion prior to disaster is a strong predictor of recovery (Jewett et al., 2021). This review corroborates this wider finding by illustrating how social support and social cohesion can facilitate the effectiveness of MHPSS interventions. Enabling social networks, such as through organising camps around existing community structures, can

relinquish the power of the community and strengthen the MHPSS response to disasters. This finding also has important implications for community engagement in MHPSS and participatory programme design.

This review brings newfound attention to natural disaster-specific barriers and facilitators MHPSS and this is a major strength. Differentiation between natural disasters and conflict is omitted from current international guidance and policy, with the IASC and the World Health Organisation often conflating natural disasters and conflict into the catch-all term 'emergency settings' (IASC, 2007; WHO, 2021). The literature sheds light on natural disaster-specific considerations including both physical barriers and emotional barriers related to delivering MHPSS following natural disasters specifically. Although it is acknowledged that natural disasters and conflict are not always mutually exclusive, these findings provide a basis for further research in this area.

Critically, further research on MHPSS in relation to both natural disasters and the developing country context is needed. Africa is one of the most vulnerable continents to the impacts of climate change, especially drought, and yet only two African studies were eligible for this review and neither focused on drought as an issue (IPCC, 2007). The authors included in this systematic review also highlighted the paucity of literature in this space in the Asian context- China will soon be classified by the World Bank as a higher-income country, and studies focused on China accounted for over half of the Asian studies in this review (Xuanmin, 2022). The methodology in this systematic review was also dominated by cross-sectional survey studies which may also reflect the general difficulty of conducting research in disaster settings (Mezinska et al., 2016).

In the current review, maximising the sustainability of the intervention through supporting local mental health infrastructure was a key facilitator to the long-term success of MHPSS. This mirrors similar findings in HICs and the frequently cited dilemma of the humanitarian-development nexus (Ando et al., 2017; Strand, 2020). However, there is a greater imbalance between government mental health spending and mental health disease burden in developing countries compared to developed countries (ranging from 3:1 in Canada to 435:1 in Haiti) (Vigo et al., 2019). While the WHA target for 80% of countries to have a system in place for MHPSS in emergencies is a key step forward, the implementation of this target in the context of other national and international priorities remains to be seen (WHO, 2021). Indeed, research in HIC settings overwhelmingly points to barriers with the utilisation and coordination of existing clinical services rather than the creation of new services and task-sharing programmes that have dominated this review, perhaps reflecting the vastly different socioeconomic contexts (Witteveen et al., 2012; Satinsky et al., 2019).

### Strengths and limitations

Single screening of abstracts was undertaken; however, this was necessary due to the solo undertaking of the project and numerous relevant databases were searched to mitigate against accidental exclusion of relevant studies. For a similar rationale, limited search terms were used, for example, focusing on disaster victim instead of providers too, and focusing on mental health conditions instead of mental health more broadly. This may have limited the number of titles and abstracts retrieved. The MHPSS interventions included were skewed towards specialised interventions meaning that the findings of this review may be less applicable to interventions at the bottom of the MHPSS pyramid. This review overcomes

methodological critiques of previous similar reviews because it clearly defines what is meant by 'barrier' and 'facilitator', outlines a clear approach to synthesis and analysis, and engages critically with the reliability of the factors identified (Bach-Mortensen and Verboom, 2020).

### Conclusion

MHPSS programmes in developing countries following natural disasters should incorporate local participation and proactive engagement with marginalised communities, build social networks, normalise mental health responses to natural disasters, strengthen local mental health infrastructure and adapt to natural disaster-specific barriers to delivery. MHPSS disaster policies should focus on decolonialising existing guidance, building resilience through task sharing approaches and establishing long-term funding streams for mental health. Ultimately, with the high likelihood of increasingly severe and frequent natural disasters, disproportionately affecting the Global South, the need to further the academic literature on MHPSS interventions in this context and build disaster preparedness in relation to mental health is imperative.

**Open peer review.** To view the open peer review materials for this article, please visit http://doi.org/10.1017/gmh.2023.91.

**Supplementary Materials.** The supplementary material for this article can be found at http://doi.org/10.1017/gmh.2023.91.

**Acknowledgement.** The authors are grateful to Dr Alexandra Conseil for her guidance on this systematic review, and to the LSHTM library staff for supporting us with the search strategy.

**Data availability statement.** The original data extraction table used in this study is available from O.R. on reasonable request. All information used to generate the thematic analysis are from the publicly available studies listed in this paper.

**Author contribution.** O.R. conceptualised the study and developed the review methodology under the supervision of A.N. O.R. ran the search strategy, identified eligible papers, extracted and curated the data and conducted the formal analysis. O.R. prepared the original draft of the manuscript and made subsequent revisions. A.N. reviewed and edited the original manuscript and its subsequent versions. Both authors approved the final version of the manuscript before submission.

**Competing interest.** The authors declare none.

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
