## [Reviewer Report]

Peer review feedback

ABSTRACT:

Background:

- It might be helpful to highlight the increase in natural disasters in developing countries due to climate change and the importance of providing a mental health response in the context of natural disasters, rather than the paucity of research in this area compared to war and conflict. Particularly given the limited availability of words here and capturing the context of the review. While I do agree the two can get conflated under the term ‘humanitarian’ response, I am uncertain if there is a paucity of research in this area, or just simply less research compared to other types of humanitarian crises.

Methods

- It would be helpful to delete the list of databases (again due to word count) and include the eligibility criteria, and quality appraisal / MMAT Tool

Results

- I would avoid selective reporting of descriptive findings (e.g., China and Earthquakes) and focus on the key themes identified.

Conclusion:

- Unclear at this stage why more research is needed. The size of an evidence-base is often a subjective judgement; however, you could argue that thirty-seven includes for a question on implementation is relatively substantial.

BACKGROUND

- Minor point, but slightly unclear what is meant by the bidirectional relationship between poverty and mental illness (Lund et al., 2011):

- Can you provide the page number for the quote: Mental health and psychosocial support (MHPSS), defined as “any type of local or 65 outside support that aims to protect or promote psychosocial well-being or prevent or 66 treat mental disorders” (UNHCR, 2021)

- Lines 70-75: It would be helpful if the statements about the effectiveness of MHPSS programmes were specific to natural disasters or highlighted the lack of disaggregation at this point.

- Lines 87-96; you make arguable points about the different mental health responses people may have in the context of a natural disaster compared to violence and war. However, it leads on from a point about gaps in the literature on implementation. When considering the aims of this review, as a reader, I would like to know why is it important that we understand the barriers and facilitators of implementing MHPSS in the context of natural disasters?

- Line 101-102: unclear what inequities means here. If it means increasing access to appropriate/high quality/affordable/suitable mental health and psychosocial support, it would be great if these issues were explored and highlighted earlier.

METHODS

- The eligibility criteria are largely transparent. However, it would be easier to follow if the definitions preceded or were reported alongside the criteria. This could be in a table or with some headings (e.g., population, intervention, humanitarian context, geographical location study design, etc). The language and no date limit reported could be reported last.

- Did you apply a definition of depression, anxiety, and PTSD? Or include all studies using these terms as determined by the study authors? Did you include/exclude PTSS? (Further details in the appendices would be great if word count is limited).

- Line 147: can you include the PRISMA checklist as part of the supplementary material?

- Can you use the PRISMA reference (e.g., Moher et al.) rather than Troup et al.?

- Line 165-172: What descriptive information did you collect via the data extraction form/tool? Were any quality assurance steps taken?

RESULTS

- Figure 1: Please check the numbers. 3285 minus 1037 =2248. Maybe the 2300 records screened includes citations hand searched? These can be included in the flow diagram

- It is usually customary to provide a breakdown of which criteria have been applied at full text, rather than composite number as this supports transparency in reporting.

- Did you also include linked studies? (e.g., where there is more than one paper reporting on a single study)

- Table 1:

o unsure what ‘Mix of MHPSS approaches’ means for Chung? Did they investigate MHPSS defined as ‘mixed’ or did they consider different types of MHPSS?

o Does Doocy’s cash for work programme meet your criteria for MHPSS?

o Do the 14 studies investigating ‘general mental health’ meet your depression, anxiety and PTSD criteria?

o Jha, some text is missing: cognitive behaviour therapy and?

o The majority of the studies include samples of survivors/recipients of MHPSS rather than providers, and I am wondering if this is because of the search strategy not including terms for providers? Was this the case for all the databases searched?

o Would it be possible to include the date of the natural disaster to better understand the timing of the study with the timing of the disaster, this could come after or merged with country of disaster to provide context details first.

- Line 236: the IASC tiers/pyramid has not been mentioned before. If you have this applied this to the studies you could mention this in the methods and the detail it could be included as a diagram or in the appendices.

- The quality of the findings based on the MMAT tool have not been included. Can this be provided in the main text of supplementary material.

- Overall, it is difficult to assess if the findings are grounded in the data, as they are not supported by participant quotes or author descriptions.

- It is unclear how many studies generated / contributed to each theme.

- Considering both of the points above, it is difficult to ascertain if findings from all studies contributed to the synthesis.

- I find the reporting of barriers and facilitators covering the same theme but reported separately/much further down in the text difficult to follow. An overview of the findings in the table would be helpful. This could include the summary of the themes, relevant quotes/author description, which studies contributed to each theme, and the quality of the studies (for each theme) would really aid the transparency in the reporting of findings.

- Line: 245: The use of some is a bit vague. Did you code for how many relied on WPP or whether studies did or did not adapt their programmes to be culturally sensitive/relevant?

- Line 248: Unclear what is meant by ‘ineffective’ here? Is that based on recipient perspectives or quantitative measures of effect? If the latter, that would be better assessed using meta-analysis, rather than thematic analysis.

DISCUSSION

- Line 415-417: Given my previous point about the extent to which you have explored cultural adaption of programmes, you could possibly make a claim about context, adaptation, and implementation. I would argue that making claims about context, and effectiveness would require a different methodological approach to the one taken in your review.

- Line 419-426: I cannot remember the extent to which the IASC guidelines engage with the issues of WPP, but there is on-going debate in the MHPSS community about how they are approaching this.

- Line 423: I would argue, MHPSS in the broadest sense, not just psychotherapy

- There is a slight overreliance on reference to the IASC guidelines, did you consider any recent guidelines issued in the light of COVID-19 that might be relevant to more recent debates on equity, accessibility etc?

- Line 476-483: I will politely disagree here. I think the findings on cultural relevance, task sharing, and stigma are common to many emergencies and even non-emergency / low-resource settings. Some of the findings on physical barriers could be argued, to speak to the timing and/or protracted nature of a given disaster. Not to say there are not important differences, but when designing and delivering the intervention, similar factors related to relevance, resource, accessibility, accessibility, stigma, etc, remain.

Strength and limitations

- I am a bit surprised that only N=3285 titles and abstracts were found, considering the lack of date on the search. However, this may be due to only searching three databases (Embase, Medline and psychinfo), the limited terms used for disaster victim (e.g., not including providers) as well as narrowing on outcomes. I recommend including this in the strengths and limitations.

CONCLUSION

- No comments

---

## [Reviewer Report]

This paper explores an interesting issue, which would be of value to the MHPSS field. The Abstract is very clear and concise, and gives the reader a good understanding of the content of the paper. The first three paragraphs of the intervention provide a clear rationale for the paper, and include appropriate and helpful references.

However, the authors’ use of key MHPSS concepts and resources is not always as clear and appropriate, and does not reflect the deep understanding of the field that would form a more solid foundation for this paper. For example, in para 4 of the Background section the definition of MHPSS is attributed to a 2021 UNHCR publication, whereas in fact MHPSS was first defined in this way in the 2007 IASC MHPSS Guidelines, and this definition has been adopted by others since then. It would be more helpful to ground the discussion in the original document, since it is foundational in the field.

The following paragraph (bottom of p3) gives an unrepresentative description of MHPSS programmes. Psychotherapy services are actually a very small aspect of MHPSS programming, and whilst the provision of basic services are part of MHPSS it is more about the extent to which these services are provided in ways which promote good mental health and psychosocial wellbeing. The MHPSS field has made great efforts to shift the focus away from PTSD and severe mental health disorders, to recognise that these affect a minority of the population. The MHPSS intervention pyramid (from the IASC MHPSS Guidelines) would provide a better overview of MHPSS programming.

A more general concern, related to this, is that the systematic review focuses on a very narrow set of outcomes (depression, anxiety and PTSD) which are not, in fact, the target outcomes of the majority of MHPSS programmes (see the recent IASC Common M&E Framework for MHPSS Programming in Emergencies for evidence of this). The paper needs to make much clearer that it is not, in fact, focusing on MHPSS programming in general, but on programming which aims to alleviate symptoms of mental disorder in contexts of disaster.

The MHPSS intervention pyramid would form a useful framework to clarify this point, and to inform the analysis. A review of Table 1 makes clear that almost all the programmes reviewed were at Level 3 of the intervention pyramid, with some Level 2 programmes represented. It is important to make this relatively narrow focus clearer, perhaps with a column in the table to indicate the level of the pyramid which each programme is positioned within. This would enable the authors to draw more meaningful conclusions about barriers and facilitators for certain types of programmes. In particular, the description of programmes in the ‘study characteristics’ section (lines 235-239) could be usefully structured according to the levels of the pyramid, with a brief description of the types of programmes represented at each level.

The paper gives the impression in places that most MHPSS programming takes place in high-resource countries, which is not in fact the case. MHPSS programming is focused primarily on emergency settings, which are more prevalent in low-resource settings. Whilst research is more often conducted in high-resource settings, as the authors point out, it is important to make clear that published research in the MHPSS field does not reflect practice. See, for example, Tol et al (2011) Mental health and psychosocial support in humanitarian settings: linking practice and research. Lancet. 2011;378(9802):1581–91 (a rather old paper but good starting point – I’m not sure things have changed greatly).

The analysis of the facilitators and barriers is helpful and well-grounded in the papers reviewed. It could be strengthened if the authors engaged more fully with recent literature and developments in the MHPSS field (e.g. WHO Quality Rights initiative and the many many papers which now exist around task-sharing in relation to MHPSS programming). The focus on reduced mental health disorder symptoms as outcomes sometimes leads the authors to imply that populations who have survived disaster always need access to mental health services/ specialists (e.g. lines 393-398), whereas there is a strong body of evidence that this is not the case, and for the majority of people traditional and community supports will be most useful and appropriate. Again, the use of the MHPSS intervention pyramid as a framework would help to make this clear.

There are some more serious misunderstandings regarding the MHPSS field, such as the statement that MHPSS programming is primarily psychotherapy which is imported from western contexts (e.g. lines 422-426). This is the opposite, in fact, of recommendations for MHPSS programming – a review of the 2007 IASC MHPSS Guidelines and the more recent Minimum Standards in MHPSS programming would make this clear.

A minor point relates to referencing style, which is sometimes inconsistent, and typing errors. The paper should be proofread carefully before being resubmitted.

Overall, this paper certainly has a useful contribution to make to the field. However, it needs to be more solidly grounded in MHPSS literature and resources, and to make clear that it is focusing on a rather narrow aspect of MHPSS programming rather than the field as a whole. The MHPSS intervention pyramid would form a useful framework for the paper.

---

## [Reviewer Report]

Thank you for giving me the opportunity to review this article. It is a valuable contribution to the literature. Please see a few comments before:

1. There has been a recent debate about the use of ‘natural’ when referring to disasters. I suggest using extreme weather events instead. You can read more here: https://reliefweb.int/report/world/why-disasters-are-not-natural.

2. I suggest strengthening the intro and discussing the mental health and psychosocial impacts and the role of MHPSS in the aftermath of extreme weather events. Why is this important and what role does it play in the response?

3. I suggest using in-text citations under study characteristics when referring to the n=.

4. Under cultural relevance, I suggest diving slightly deeper into this concept of western therapeutic modalities being used without proper adaptation and contextualization. Say more about why this is a barrier to delivery and the effectiveness of the intervention (it is not just a barrier to delivery but to effectiveness). This speaks to the need for DRR in counteries where extreme weather events are common. This is mentioned in line 437 but I suggest building this out more as a significant barrier highlighted in your paper. What can be done to prepare frontline workers, mental health, protection, communities themselves, etc. to prepare for extreme weather events and have the skills and tools to support one another. This is often one of the biggest barriers, services simply not existing.

5. Line 261-267, basic mental health and psychosocial considerations, activities and interventions are often neglected, not just the interventions. When you refer to social work for mental health, what does this mean? Are you referring to case management? This should be clarified and explained further.

6. Line 322, I suggest using a different term than trauma issues, such as distress. I don’t believe the term trauma issue is utilized or will have meaning to the reader.

7. Line 269, I suggest adding more about the misconception surrounding only mental health professionals delivering quality mental health care. You dive into this in the discussion but it can also be highlighted here. Besides the lack of mental health staff, is there all a belief that only mental health professionals can provide these services or that in the aftermath of an extreme weather event, mental health care is not a priority?

8. Stigma is a huge barrier to the uptake of mental health services. I suggest saying more about how stigma is a barrier to MHPSS services following a disaster.

9. The paper focuses heavily on trauma. I suggest using terminology such as distress and adverse as there is a shift away from trauma and PTSD being the sole focus of response as it lends itself to a deficit model. Line 58, for example, I suggest saying, risk factors include exposure to the event.

10. On line 469, you highlight the role of social support as a factor that influences mental health and wellbeing. I suggest diving deeper here and discussing the role of community engagement and support in the aftermath of extreme weather events. I also suggest discussing how critical social cohesion and collective action is to support recovery efforts and ultimately the wellbeing of the community.

---

## [Reviewer Report]

Comments revisor 1: 

ABSTRACT:

Background:

- It might be helpful to highlight the increase in natural disasters in developing countries due to climate change and the importance of providing a mental health response in the context of natural disasters, rather than the paucity of research in this area compared to war and conflict. Particularly given the limited availability of words here and capturing the context of the review. While I do agree the two can get conflated under the term ‘humanitarian’ response, I am uncertain if there is a paucity of research in this area, or just simply less research compared to other types of humanitarian crises.

Methods

- It would be helpful to delete the list of databases (again due to word count) and include the eligibility criteria, and quality appraisal / MMAT Tool

Results

- I would avoid selective reporting of descriptive findings (e.g., China and Earthquakes) and focus on the key themes identified.

Conclusion:

- Unclear at this stage why more research is needed. The size of an evidence-base is often a subjective judgement; however, you could argue that thirty-seven includes for a question on implementation is relatively substantial.

BACKGROUND

- Minor point, but slightly unclear what is meant by the bidirectional relationship between poverty and mental illness (Lund et al., 2011):

- Can you provide the page number for the quote: Mental health and psychosocial support (MHPSS), defined as “any type of local or 65 outside support that aims to protect or promote psychosocial well-being or prevent or 66 treat mental disorders” (UNHCR, 2021)

- Lines 70-75: It would be helpful if the statements about the effectiveness of MHPSS programmes were specific to natural disasters or highlighted the lack of disaggregation at this point.

- Lines 87-96; you make arguable points about the different mental health responses people may have in the context of a natural disaster compared to violence and war. However, it leads on from a point about gaps in the literature on implementation. When considering the aims of this review, as a reader, I would like to know why is it important that we understand the barriers and facilitators of implementing MHPSS in the context of natural disasters?

- Line 101-102: unclear what inequities means here. If it means increasing access to appropriate/high quality/affordable/suitable mental health and psychosocial support, it would be great if these issues were explored and highlighted earlier.

METHODS

- The eligibility criteria are largely transparent. However, it would be easier to follow if the definitions preceded or were reported alongside the criteria. This could be in a table or with some headings (e.g., population, intervention, humanitarian context, geographical location study design, etc). The language and no date limit reported could be reported last.

- Did you apply a definition of depression, anxiety, and PTSD? Or include all studies using these terms as determined by the study authors? Did you include/exclude PTSS? (Further details in the appendices would be great if word count is limited).

- Line 147: can you include the PRISMA checklist as part of the supplementary material?

- Can you use the PRISMA reference (e.g., Moher et al.) rather than Troup et al.?

- Line 165-172: What descriptive information did you collect via the data extraction form/tool? Were any quality assurance steps taken?

RESULTS

- Figure 1: Please check the numbers. 3285 minus 1037 =2248. Maybe the 2300 records screened includes citations hand searched? These can be included in the flow diagram

- It is usually customary to provide a breakdown of which criteria have been applied at full text, rather than composite number as this supports transparency in reporting.

- Did you also include linked studies? (e.g., where there is more than one paper reporting on a single study)

- Table 1:

o unsure what ‘Mix of MHPSS approaches’ means for Chung? Did they investigate MHPSS defined as ‘mixed’ or did they consider different types of MHPSS?

o Does Doocy’s cash for work programme meet your criteria for MHPSS?

o Do the 14 studies investigating ‘general mental health’ meet your depression, anxiety and PTSD criteria?

o Jha, some text is missing: cognitive behaviour therapy and?

o The majority of the studies include samples of survivors/recipients of MHPSS rather than providers, and I am wondering if this is because of the search strategy not including terms for providers? Was this the case for all the databases searched?

o Would it be possible to include the date of the natural disaster to better understand the timing of the study with the timing of the disaster, this could come after or merged with country of disaster to provide context details first.

- Line 236: the IASC tiers/pyramid has not been mentioned before. If you have this applied this to the studies you could mention this in the methods and the detail it could be included as a diagram or in the appendices.

- The quality of the findings based on the MMAT tool have not been included. Can this be provided in the main text of supplementary material.

- Overall, it is difficult to assess if the findings are grounded in the data, as they are not supported by participant quotes or author descriptions.

- It is unclear how many studies generated / contributed to each theme.

- Considering both of the points above, it is difficult to ascertain if findings from all studies contributed to the synthesis.

- I find the reporting of barriers and facilitators covering the same theme but reported separately/much further down in the text difficult to follow. An overview of the findings in the table would be helpful. This could include the summary of the themes, relevant quotes/author description, which studies contributed to each theme, and the quality of the studies (for each theme) would really aid the transparency in the reporting of findings.

- Line: 245: The use of some is a bit vague. Did you code for how many relied on WPP or whether studies did or did not adapt their programmes to be culturally sensitive/relevant?

- Line 248: Unclear what is meant by ‘ineffective’ here? Is that based on recipient perspectives or quantitative measures of effect? If the latter, that would be better assessed using meta-analysis, rather than thematic analysis.

DISCUSSION

- Line 415-417: Given my previous point about the extent to which you have explored cultural adaption of programmes, you could possibly make a claim about context, adaptation, and implementation. I would argue that making claims about context, and effectiveness would require a different methodological approach to the one taken in your review.

- Line 419-426: I cannot remember the extent to which the IASC guidelines engage with the issues of WPP, but there is on-going debate in the MHPSS community about how they are approaching this.

- Line 423: I would argue, MHPSS in the broadest sense, not just psychotherapy

- There is a slight overreliance on reference to the IASC guidelines, did you consider any recent guidelines issued in the light of COVID-19 that might be relevant to more recent debates on equity, accessibility etc?

- Line 476-483: I will politely disagree here. I think the findings on cultural relevance, task sharing, and stigma are common to many emergencies and even non-emergency / low-resource settings. Some of the findings on physical barriers could be argued, to speak to the timing and/or protracted nature of a given disaster. Not to say there are not important differences, but when designing and delivering the intervention, similar factors related to relevance, resource, accessibility, accessibility, stigma, etc, remain.

Strength and limitations

- I am a bit surprised that only N=3285 titles and abstracts were found, considering the lack of date on the search. However, this may be due to only searching three databases (Embase, Medline and psychinfo), the limited terms used for disaster victim (e.g., not including providers) as well as narrowing on outcomes. I recommend including this in the strengths and limitations.

Comments revisor 2: This paper explores an interesting issue, which would be of value to the MHPSS field. However, the authors’ use of key MHPSS concepts and resources is not always as clear and appropriate, and does not reflect the deep understanding of the field that would form a more solid foundation for this paper. For example, in para 4 of the Background section the definition of MHPSS is attributed to a 2021 UNHCR publication, whereas in fact MHPSS was first defined in this way in the 2007 IASC MHPSS Guidelines, and this definition has been adopted by others since then. It would be more helpful to ground the discussion in the original document, since it is foundational in the field.

The following paragraph (bottom of p3) gives an unrepresentative description of MHPSS programmes. Psychotherapy services are actually a very small aspect of MHPSS programming, and whilst the provision of basic services are part of MHPSS it is more about the extent to which these services are provided in ways which promote good mental health and psychosocial wellbeing. The MHPSS field has made great efforts to shift the focus away from PTSD and severe mental health disorders, to recognise that these affect a minority of the population. The MHPSS intervention pyramid (from the IASC MHPSS Guidelines) would provide a better overview of MHPSS programming.

A more general concern, related to this, is that the systematic review focuses on a very narrow set of outcomes (depression, anxiety and PTSD) which are not, in fact, the target outcomes of the majority of MHPSS programmes (see the recent IASC Common M&E Framework for MHPSS Programming in Emergencies for evidence of this). The paper needs to make much clearer that it is not, in fact, focusing on MHPSS programming in general, but on programming which aims to alleviate symptoms of mental disorder in contexts of disaster.

The MHPSS intervention pyramid would form a useful framework to clarify this point, and to inform the analysis. A review of Table 1 makes clear that almost all the programmes reviewed were at Level 3 of the intervention pyramid, with some Level 2 programmes represented. It is important to make this relatively narrow focus clearer, perhaps with a column in the table to indicate the level of the pyramid which each programme is positioned within. This would enable the authors to draw more meaningful conclusions about barriers and facilitators for certain types of programmes. In particular, the description of programmes in the ‘study characteristics’ section (lines 235-239) could be usefully structured according to the levels of the pyramid, with a brief description of the types of programmes represented at each level.

The paper gives the impression in places that most MHPSS programming takes place in high-resource countries, which is not in fact the case. MHPSS programming is focused primarily on emergency settings, which are more prevalent in low-resource settings. Whilst research is more often conducted in high-resource settings, as the authors point out, it is important to make clear that published research in the MHPSS field does not reflect practice. See, for example, Tol et al (2011) Mental health and psychosocial support in humanitarian settings: linking practice and research. Lancet. 2011;378(9802):1581–91 (a rather old paper but good starting point – I’m not sure things have changed greatly).

The analysis of the facilitators and barriers is helpful and well-grounded in the papers reviewed. It could be strengthened if the authors engaged more fully with recent literature and developments in the MHPSS field (e.g. WHO Quality Rights initiative and the many many papers which now exist around task-sharing in relation to MHPSS programming). The focus on reduced mental health disorder symptoms as outcomes sometimes leads the authors to imply that populations who have survived disaster always need access to mental health services/ specialists (e.g. lines 393-398), whereas there is a strong body of evidence that this is not the case, and for the majority of people traditional and community supports will be most useful and appropriate. Again, the use of the MHPSS intervention pyramid as a framework would help to make this clear.

There are some more serious misunderstandings regarding the MHPSS field, such as the statement that MHPSS programming is primarily psychotherapy which is imported from western contexts (e.g. lines 422-426). This is the opposite, in fact, of recommendations for MHPSS programming – a review of the 2007 IASC MHPSS Guidelines and the more recent Minimum Standards in MHPSS programming would make this clear.

A minor point relates to referencing style, which is sometimes inconsistent, and typing errors. The paper should be proofread carefully before being resubmitted.

Overall, this paper certainly has a useful contribution to make to the field. However, it needs to be more solidly grounded in MHPSS literature and resources, and to make clear that it is focusing on a rather narrow aspect of MHPSS programming rather than the field as a whole. The MHPSS intervention pyramid would form a useful framework for the paper. 

Comments revisor 3: Please see a few comments before:

1. There has been a recent debate about the use of ‘natural’ when referring to disasters. I suggest using extreme weather events instead. You can read more here: https://reliefweb.int/report/world/why-disasters-are-not-natural.

2. I suggest strengthening the intro and discussing the mental health and psychosocial impacts and the role of MHPSS in the aftermath of extreme weather events. Why is this important and what role does it play in the response?

3. I suggest using in-text citations under study characteristics when referring to the n=.

4. Under cultural relevance, I suggest diving slightly deeper into this concept of western therapeutic modalities being used without proper adaptation and contextualization. Say more about why this is a barrier to delivery and the effectiveness of the intervention (it is not just a barrier to delivery but to effectiveness). This speaks to the need for DRR in counteries where extreme weather events are common. This is mentioned in line 437 but I suggest building this out more as a significant barrier highlighted in your paper. What can be done to prepare frontline workers, mental health, protection, communities themselves, etc. to prepare for extreme weather events and have the skills and tools to support one another. This is often one of the biggest barriers, services simply not existing.

5. Line 261-267, basic mental health and psychosocial considerations, activities and interventions are often neglected, not just the interventions. When you refer to social work for mental health, what does this mean? Are you referring to case management? This should be clarified and explained further.

6. Line 322, I suggest using a different term than trauma issues, such as distress. I don’t believe the term trauma issue is utilized or will have meaning to the reader.

7. Line 269, I suggest adding more about the misconception surrounding only mental health professionals delivering quality mental health care. You dive into this in the discussion but it can also be highlighted here. Besides the lack of mental health staff, is there all a belief that only mental health professionals can provide these services or that in the aftermath of an extreme weather event, mental health care is not a priority?

8. Stigma is a huge barrier to the uptake of mental health services. I suggest saying more about how stigma is a barrier to MHPSS services following a disaster.

9. The paper focuses heavily on trauma. I suggest using terminology such as distress and adverse as there is a shift away from trauma and PTSD being the sole focus of response as it lends itself to a deficit model. Line 58, for example, I suggest saying, risk factors include exposure to the event.

10. On line 469, you highlight the role of social support as a factor that influences mental health and wellbeing. I suggest diving deeper here and discussing the role of community engagement and support in the aftermath of extreme weather events. I also suggest discussing how critical social cohesion and collective action is to support recovery efforts and ultimately the wellbeing of the community.

---

## [Reviewer Report]

Dear Dr Bass

Thank you for giving us the opportunity to respond to the reviewers' feedback and to make relevant changes based on their input. We have given a detailed response to the feedback and also revised the manuscript based on the feedback.

We look forward to your decision based on our response and revised manuscript.

Best wishes

Abhijit

---

## [Reviewer Report]

I appreciate the efforts made by the authors to address the issues I raised in my earlier review. In my opinion, the paper is now much stronger and almost ready for publication. I have one small suggested revision, described below.

I like the inclusion of ‘tiers’ in Table 1 to clarify the level of MHPSS intervention addressed by the programmes evaluated in the selected papers. I would say, though, that the Goenjian (2021) and the James (2013) papers actually describe a Tier 3 intervention – Tier 2 interventions are about strengthening social supports and would be open to all within the target group, not only those with high levels of distress (Tier 1 and 2 are primarily about promoting good psychosocial wellbeing and preventing mental health problems). A group psychotherapy intervention would be Tier 3 or 4. This might also apply to the Gao (2013) paper and the Ho (2017) papers, depending on whether these interventions targeted individuals with high levels of distress, or were open to all.

If the authors made these revisions, this would have implications for the first para of the ‘study characteristics’ section on p17.

Other than this, I’m happy to recommend publication for this revised version of the paper. I’ve enjoyed reviewing it, thankyou for the opportunity.

---

## [Reviewer Report]

The comments I made were adequately addressed in the revision and the article is reading very well. One small comments below:

1. Page 18, line 17-18. The MHPSS pyramid is referred to as layers not tiers. For example, you can say “followed by layer 2 family and community support”. You can refer to the UNICEF Guidelines https://www.unicef.org/media/109086/file/Global%20multisectorial%20operational%20framework.pdf.

---

## [Reviewer Report]

The reviewers appreciate the efforts made by the authors to address this interesting topic. The reviewers have recommended some minor queries, which are described below.

Author 1:

I like the inclusion of ‘tiers’ in Table 1 to clarify the level of MHPSS intervention addressed by the programmes evaluated in the selected papers. I would say, though, that the Goenjian (2021) and the James (2013) papers actually describe a Tier 3 intervention – Tier 2 interventions are about strengthening social supports and would be open to all within the target group, not only those with high levels of distress (Tier 1 and 2 are primarily about promoting good psychosocial wellbeing and preventing mental health problems). A group psychotherapy intervention would be Tier 3 or 4. This might also apply to the Gao (2013) paper and the Ho (2017) papers, depending on whether these interventions targeted individuals with high levels of distress, or were open to all.

If the authors made these revisions, this would have implications for the first para of the ‘study characteristics’ section on p17.

Autho2:

The comments I made were adequately addressed in the revision and the article is reading very well. One small comments below:

1. Page 18, line 17-18. The MHPSS pyramid is referred to as layers not tiers. For example, you can say “followed by layer 2 family and community support”. You can refer to the UNICEF Guidelines https://www.unicef.org/media/109086/file/Global%20multisectorial%20operational%20framework.pdf.

---

## [Reviewer Report]

The minor points I highlighted in my second review have been addressed, and I am happy to recommend publication.

---

## [Reviewer Report]

The paper has come along way and it has been a pleasure reviewing!

1. Page 3, Line 41, Ii looks like there is a misplaced period here after and. indirect.

2. Page 4, Line 19, “Most people will be able to recover from experiencing a disaster through basic MHPSS services like the provision of shelter, food and community support, however a minority of individuals will require more specialist mental health interventions to cope (DeWolfe et al., 2000). It should read, ”will require more focused or specialized care...". This is per the IASC MHPSS intervention pyramid.

3. Page 4, line 28. “MHPSS programmes, both psychotherapy services and basic services, have been found to be effective in improving mental health outcomes in individuals affected by humanitarian emergencies in developing countries, including by improving psychological functioning and reducing the prevalence of post-traumatic stress disorders (Bangpan et al., 2019)”. I suggest referring to the IASC pyramid layers here, “...MHPSS programmes including basic services, community supports and focused care,...”.

4. P 4, line 52. The term mental distress is not commonly used. I suggest changing to psychological distress.

5. Page 7, Focussed non-speacialised support. Should be, “Focused care...”.

6. Page 11. The Berliner study. There has been a lot of debate on which later PFA sits and in conclusion, it is not an intervention but a skill set that is taught to frontline staff and volunteers. I suggest putting this at layer 1 as it is a basic service and not an intervention and definitely not focused care.

---

## [Reviewer Report]

Dear Authors, The paper has come along way and it has been a pleasure review the next comments:

1. Page 3, Line 41, Ii looks like there is a misplaced period here after and. indirect.

2. Page 4, Line 19, “Most people will be able to recover from experiencing a disaster through basic MHPSS services like the provision of shelter, food and community support, however a minority of individuals will require more specialist mental health interventions to cope (DeWolfe et al., 2000). It should read, ”will require more focused or specialized care...". This is per the IASC MHPSS intervention pyramid.

3. Page 4, line 28. “MHPSS programmes, both psychotherapy services and basic services, have been found to be effective in improving mental health outcomes in individuals affected by humanitarian emergencies in developing countries, including by improving psychological functioning and reducing the prevalence of post-traumatic stress disorders (Bangpan et al., 2019)”. I suggest referring to the IASC pyramid layers here, “...MHPSS programmes including basic services, community supports and focused care,...”.

4. P 4, line 52. The term mental distress is not commonly used. I suggest changing to psychological distress.

5. Page 7, Focussed non-speacialised support. Should be, “Focused care...”.

6. Page 11. The Berliner study. There has been a lot of debate on which later PFA sits and in conclusion, it is not an intervention but a skill set that is taught to frontline staff and volunteers. I suggest putting this at layer 1 as it is a basic service and not an intervention and definitely not focused care.